# Towards Conditionally Dependent Masked Language Models

## Abstract

Masked language modeling has proven to be an effective paradigm for learning representations of language. However, when multiple tokens are masked out, the masked language model's (MLM) distribution over the masked positions assumes that the masked tokens are conditionally independent given the unmasked tokens—an assumption that does not hold in practice. Existing work addresses this limitation by interpreting the sum of unary scores (i.e., the logits or the log probabilities of single tokens when conditioned on all others) as the log potential a Markov random field (MRF). While this new model no longer makes any independence assumptions, it remains unclear whether this approach (i) results in a good probabilistic model of language and further (ii) derives a model that is faithful (i.e., has matching unary distributions) to the original model. This paper studies MRFs derived this way in a controlled setting where only two tokens are masked out at a time, which makes it possible to compute exact distributional properties. We find that such pairwise MRFs are often worse probabilistic models of language from a perplexity standpoint, and moreover have unary distributions that do not match the unary distributions of the original MLM. We then study a statistically-motivated iterative optimization algorithm for deriving joint pairwise distributions that are more compatible with the original unary distributions. While this iterative approach outperforms the MRF approach, the algorithm itself is too expensive to be practical. We thus amortize this optimization process through a parameterized feed-forward layer that learns to modify the original MLM's pairwise distributions to be both non-independent and faithful, and find that this approach outperforms the MLM for scoring pairwise tokens.

## 1 Introduction

Masked language modeling has proven to be an effective paradigm for learning generalizable representations of language (Devlin et al., 2019; Liu et al., 2019; He et al., 2021) and other structured domains (Rives et al., 2021; Mahmood et al., 2021; He et al., 2022). From a probabilistic perspective, masked language models (MLM) make strong independence assumptions. When multiple tokens are masked out, MLMs assume that the distributions over the masked tokens are conditionally independent given the unmasked tokens—an assumption that clearly does not hold for language. For example, consider the sentence: "The [MASK]$_1$ [MASK]$_2$ pleasantly surprised by an analysis paper." MLM's assume that the distribution over the two tokens are independent and thus cannot systematically assign higher probability to grammatical subject-verb agreements ("reviewer was" and "reviewers were") than ungrammatical ones (*"reviewer were" and *"reviewers was"). These types of statistical dependencies can occur for words that are far apart,

> The [MASK]$_1$, tired from reading so many papers that focused on performance gains, [MASK]$_2$ pleasantly surprised by an analysis paper.

Indeed, such long-range dependencies animate much work on hierarchical approaches to language which posit (usually tree-like) structures in which words that are "close" in structure space (but potentially far apart in surface form) have high dependency with one another.

From purely a representation learning perspective, such model misspecifications arising from incorrect statistical assumptions may not be catastrophic. These assumptions can enable scalable training and even aid in learning better representations by serving as a statistical bottleneck that forces more

information to be captured by the hidden states.[1] However, we observe that MLMs are increasingly being employed as *probabilistic* models of language, for example for scoring (Salazar et al., 2020; Xu et al., 2022) and sampling/decoding (Wang & Cho, 2019; Ghazvininejad et al., 2019; Ng et al., 2020; Yamakoshi et al., 2022) sentences. Under such probabilistic uses of MLMs, it becomes critical to ensure that the underlying statistical assumptions are plausibly grounded in reality.

Existing work has approached this problem by using the conditionals of an MLM to define an alternative probabilistic model of language that does not make said conditional independence assumptions. Noting that the *unary* conditional distributions of an MLM (i.e., the conditional distributions output by the MLM when a single token is masked out) do not make any independence assumptions, Goyal et al. (2022) define a fully connected Markov random field (MRF) language model whose log potential of a sentence is defined to the sum of these unary log probabilities (or logits). This approach, while sensible, raises two questions: (i) is this new model a good probabilistic model of language, and (ii) are the conditionals of the derived model *faithful* to the original MLM, i.e., are the unary conditionals of the new model the same as (or similar to) the unary conditionals of the MLM?[2] The latter faithfulness question is important because due to the scale at which these models are trained, it is not completely outrageous to posit that the unary conditionals learned by the MLM are close enough the true unary distributions of language.[3]

This paper investigates both questions in a controlled *pairwise conditional* setting where only two tokens are masked out at a time, which makes it possible to compute the MRF's pairwise distribution exactly. Surprisingly, we find that such pairwise MRFs are often a worse probabilistic model of language than even the original MLM that assumes independence between the two masked tokens. We moreover find that the MRF's unary distributions do not match the MLM's unary distributions. In light of this result, we study two alternative approaches to deriving non-independent pairwise distributions from the MLM's unary distributions. The first approach exploits the Hammersley–Clifford–Besag theorem (Besag, 1974), which allows one to write down a joint distribution in terms of unary conditionals. The second approach uses an iterative algorithm that finds a joint distribution over two masked positions whose unary conditionals are closest, in the KL sense, to the unary conditionals of the MLM (Arnold & Gokhale, 1998). We find that joint pairwise distributions from the iterative approach have better perplexity than both the MRF and the MLM, and also have unary conditionals that are closer to those of the original MLM's. While effective, the iterative algorithm is too expensive to be practical. We thus propose an amortized variant of the iterative approach that can compute non-independent pairwise conditionals using only a single forward pass of the MLM followed by an efficient feed-forward layer, and find that this amortized approach outperforms original MLM when scoring adjacent pairwise tokens. Our code will be made publicly available.

## 2 BACKGROUND

We begin by introducing notation. Let $\mathcal{V}$ be a vocabulary of tokens, and $T$ be the text length, and $\mathbf{w} \in \mathcal{V}^T$ be an input sentence/paragraph. We are particularly interested in the case when a subset $S \subseteq [T] \triangleq \{1, \ldots, T\}$ of the input $\mathbf{w}$ is replaced with the special [MASK] tokens; in this case we will use the notation $q_{t|\overline{S}}(\cdot \mid \mathbf{w}_{\overline{S}})$ to denote the output distribution of the MLM at position $t \in S$, where $\mathbf{w}_{\overline{S}}$ is derived from $\mathbf{w}$ by masking out $\mathbf{w}_t$ for all $t \in S$. MLMs are trained to maximize the log-likelihood of a set of masked words $S$ in a sentence. More formally, consider an MLM parameterized by a vector $\boldsymbol{\theta} \in \Theta$ and some distribution $\mu(\cdot)$ over subsets of positions to mask $S \subseteq [T]$. The MLM learning objective can then be written as:

$$\arg\max_{\boldsymbol{\theta}} \mathop{\mathbb{E}}_{\mathbf{w} \sim p(\cdot)} \mathop{\mathbb{E}}_{S \sim \mu(\cdot)} \left[ \frac{1}{|S|} \sum_{t \in S} \log q_{t|\overline{S}}(w_t \mid \mathbf{w}_{\overline{S}}; \boldsymbol{\theta}) \right],$$

where $p(\cdot)$ denotes the true data distribution. Let $p_{S|\overline{S}}(\cdot \mid \mathbf{w}_{\overline{S}})$ analogously be the conditionals of the data distribution and further let $q_{S|\overline{S}}(\mathbf{w}_S \mid \mathbf{w}_{\overline{S}}) \triangleq \prod_{i \in S} q_{i|\overline{S}}(w_i \mid \mathbf{w}_{\overline{S}})$ be the joint distribution

---

[1]Indeed, prior work has found that masking out contiguous words (which on average have higher dependency than non-contiguous words; Joshi et al., 2020) or employing more aggressive masking rates (Wettig et al., 2022) can improve representation learning.

[2]Of course, it is possible that the set of unary conditional distributions themselves may be *incompatible* (Arnold & Press, 1989), i.e., there is *no* joint distribution whose unary conditionals exactly equal those of the MLM's. In our empirical study we show that this is indeed the case.

[3]As noted by https://machinethoughts.wordpress.com/2019/07/14/a-consistency-theorem-for-bert/.

over the masked tokens $\mathbf{w}_S$. Then the above can be rewritten as:

$$\arg\min_{\boldsymbol{\theta}} \; \underset{S \sim \mu(\cdot)}{\mathbb{E}} \left[ \frac{1}{|S|} D_{\text{KL}}(p_{S|\overline{S}}(\cdot \mid \mathbf{w}_{\overline{S}}) \,\|\, q_{S|\overline{S}}(\cdot \mid \mathbf{w}_{\overline{S}}; \boldsymbol{\theta})) \right].$$

Thus, we can interpret MLMs learning a family of conditionals distributions $\{q_{S|\overline{S}}\}_{S \subseteq [T]}$ that assume that the masked words are conditionally independent given the unmasked words.

## 3 CONDITIONALLY DEPENDENT MASKED LANGUAGE MODELS

While the conditional independence assumption that underpins MLMs enables scalable training while conditioning on bidirectional context,[4] as noted in the introduction this assumption is clearly invalid for language, i.e., the MLM is *misspecified*. Note that all conditional distributions where we mask two or more words, i.e., when $|S| \geq 2$, are affected by this. For this reason, we will be particularly interested in the **unary conditionals** of the MLM, which arise when $|S| = 1$. If we let $S = \{t\}$ for some $t \in [T]$, we will slightly abuse notation and refer to $q_{t|\overline{t}} \triangleq q_{\{t\}|\overline{\{t\}}}$ as the unary conditional of the MLM at position $t$. Since (i) unary conditionals do not make any independence assumptions and (ii) the scale at which modern MLMs are trained is such that we might reasonably expect $D_{\text{KL}}(p_{t|\overline{t}} \,\|\, q_{t|\overline{t}})$ to be small, they become natural objects of study when trying to build conditionally *dependent* models from MLMs.

### 3.1 MARKOV RANDOM FIELDS DERIVED FROM MLMS

To address the conditional independence limitation of MLMs, recent work has proposed using the unary conditionals of the MLM to define a new probabilistic model over larger units of language such as phrases and sentences (Wang & Cho, 2019; Goyal et al., 2022). The idea is to define:

$$q^{\text{MRF}}(\mathbf{w}) \propto \prod_{t \in [T]} q_{t|\overline{t}}(w_t \mid \mathbf{w}_{\overline{t}}), \tag{1}$$

which can be interpreted as a fully connected MRF, whose log potential is given by the sum of the unary log probabilities. One can similarly define a variant of this where the log potentials are *logits* of the unary conditionals of the MLM, i.e., $q^{\text{MRF}_{\text{L}}}(\mathbf{w}) \propto \prod_{t \in [T]} s_{t|\overline{t}}(w_t \mid \mathbf{w}_{\overline{t}})$ where $s_{t|\overline{t}}(w_t \mid \mathbf{w}_{\overline{t}})$ is the logit of $w_t$ conditioned on $\mathbf{w}_{\overline{t}}$. These two models correspond to an MRF with a single fully connected clique, and thus they do not make any conditional independence assumptions.

This construction addresses the conditional independence limitation of the original MLMs, but results in a different probabilistic model, which raises two questions. First, it is not immediately clear if this probabilistic model is actually a *good* probabilistic model of language, since while we can approximately sample from this MRF with (for example) MCMC (Goyal et al., 2022), computing perplexity is completely intractable.[5] Second, despite being defined in terms of the unary conditionals, it is not clear to what extent this model is **faithful** to the original MLM, i.e., it is not clear whether the unary conditionals of the MRF are the same as (or close to) the unary conditionals of the parent MLM. This is again important because due to the scale at which these models are trained, the conditionals learned by the MLM may be close enough to the true unary distributions of language. Thus, an ideal MRF should have unary distributions that are faithful to the MLM's.

Theoretically, it is not hard to show that an MRF defined this way can have nonzero KL to the original unary distribution even in the simple two variable case where we assume that the MRF is constructed from *true* unary distributions, as given by the following preposition.

**Proposition 3.1.** *Let $w_1, w_2 \in \mathcal{V}$ and further let $p_{1|2}(\cdot \mid w_2), p_{2|1}(\cdot \mid w_1)$ be the true (i.e., population) unary conditional distributions. Define an MRF as*

$$q_{1,2}(w_1, w_2) \propto p_{1|2}(w_1 \mid w_2)\, p_{2|1}(w_2 \mid w_1),$$

*and let $q_{1|2}(\cdot \mid w_2), q_{2|1}(\cdot \mid w_1)$ be the conditionals derived from the MRF. Then there exists $p_{1|2}, p_{2|1}$ such that*

$$D_{KL}(p_{1|2}(\cdot \mid w_2) \,\|\, q_{1|2}(\cdot \mid w_2)) > 0.$$

---

[4]Though see Yang et al. (2019) for an alternative approach.

[5]The partition function of the MRF would require $T|\mathcal{V}|^{T-1}$ forward passes through the MLM. We also tried estimating the partition through importance sampling with GPT-2 (Radford et al., 2019), but found the estimate to be quite poor.

See App. A for a proof. This indicates that even in a simple setting with two variables, MRF-derived distributions may not be faithful. This motivates our pairwise setting below.

### 3.1.1 PAIRWISE MRFs

Since the partition function of the MRF is intractable, we cannot compute the unary conditionals of the MRF. We can, however, analyze the case where we use the same approach to define a *pairwise* MRF over a pair of positions. Given two positions $S = \{a, b\} \subset [T]$, and a context $\mathbf{w}_{\overline{S}}$ (i.e., a setting of all the other positions), we define a pairwise MRF over those two positions via:

$$q_{a,b|\overline{S}}^{\text{MRF}}(w_a, w_b \mid \mathbf{w}_{\overline{S}}) \propto q_{a|\overline{S}}(w_a \mid \mathbf{w}_{\overline{S}}) \, q_{b|\overline{S}}(w_b \mid \mathbf{w}_{\overline{S}})$$

The definition of $q_{a,b|\overline{S}}^{\text{MRF-L}}(\cdot \mid \mathbf{w}_{\overline{S}})$, which uses the unary conditional logits, is analogous. Doing so requires $2|\mathcal{V}|$ forward passes through the MLM for each conditional, which is expensive but tractable on modern GPUs with large batches. Given this pairwise MRF, we can now compute the perplexities and faithfulness metrics on a dataset $\mathcal{D} = \{(w_a^{(n)}, w_b^{(n)}, \mathbf{w}_{\overline{S}}^{(n)})\}_{n=1}^{N}$ of English sentences where the tokens at positions $a$ and $b$ (i.e., tokens $w_a$ and $w_b$) have been masked.

**Language model performance.** To evaluate the pairwise MRF as a language model, we compute two measures: unary perplexity and pairwise perplexity. The unary perplexity (**U-PPL**) over single tokens is given by,

$$\exp\left(-\frac{1}{2N}\sum_{n=1}^{N}\log q_{a|b,\overline{S}}^{\text{MRF}}(w_a^{(n)} \mid w_b^{(n)}, \mathbf{w}_{\overline{S}}^{(n)}) + \log q_{b|a,\overline{S}}^{\text{MRF}}(w_b^{(n)} \mid w_a^{(n)}, \mathbf{w}_{\overline{S}}^{(n)})\right).$$

An ideal model would obtain unary perplexity that is similar to the MLM's (which uses $q$ instead of $q^{\text{MRF}}$ in the above expression). The pairwise perplexity (**P-PPL**) over two tokens is given by,

$$\exp\left(-\frac{1}{2N}\sum_{n=1}^{N}\log q_{a,b|\overline{S}}^{\text{MRF}}(w_a^{(n)}, w_b^{(n)} \mid \mathbf{w}_{\overline{S}}^{(n)})\right).$$

We would expect a good model of language to obtain a lower pairwise perplexity than the original MLM which (wrongly) assumes conditional independence.

**Faithfulness.** We also use the derived unary conditionals to assess faithfulness to the MLM's unary conditionals by calculating the average conditional KL divergence (**A-KL**) between the unary conditionals,

$$\frac{1}{N|\mathcal{V}|}\sum_{n=1}^{N}\sum_{w_b \in \mathcal{V}} D_{\text{KL}}(q_{a|b,\overline{S}}(\cdot \mid w_b, \mathbf{w}_{\overline{S}}^{(n)}) \,\|\, q_{a|b,\overline{S}}^{\text{MRF}}(\cdot \mid w_b, \mathbf{w}_{\overline{S}}^{(n)})).$$

If the MRF is completely faithful to the MLM, this number should be zero. The above metric averages the KL across the entire vocabulary $\mathcal{V}$, but in practice we may only be interested in assessing closeness only when conditioned on the gold tokens. We thus compute a variant of the above metric where we only average over the conditionals for the gold token (**G-KL**):

$$\frac{1}{N}\sum_{n=1}^{N} D_{\text{KL}}(q_{a|b,\overline{S}}(\cdot \mid w_b^{(n)}, \mathbf{w}_{\overline{S}}^{(n)}) \,\|\, q_{a|b,\overline{S}}^{\text{MRF}}(\cdot \mid w_b^{(n)}, \mathbf{w}_{\overline{S}}^{(n)})).$$

This metric penalizes unfaithfulness in common contexts more than in uncommon contexts. Again, this should be zero if the models are faithful.

**Experimental setup.** We calculate the above metrics on 1000 examples from a natural language inference dataset (SNLI, Bowman et al., 2015) and a summarization dataset (XSUM, Narayan et al., 2018). Since dependencies are more likely to emerge when the tokens being masked are next to each other, we consider two schemes for selecting the two tokens to be masked for each sentence: masks over two tokens chosen uniformly at random (**Random pairs**), and also over random *contiguous* tokens (chosen uniformly at random) in a sentence (**Contiguous pairs**). For exact comparison we make sure the masking is the same for all models. In addition, we consider both BERT$_{\text{BASE}}$ and BERT$_{\text{LARGE}}$ (cased) as the MLMs from which to obtain the unary conditionals.[6]

---

[6] We use the Huggingface (Wolf et al., 2020) implementations of these models.

| | Dataset | Scheme | Random pairs | | | | Contiguous pairs | | | |
|---|---|---|---|---|---|---|---|---|---|---|
| | | | U-PPL | P-PPL | A-KL | G-KL | U-PPL | P-PPL | A-KL | G-KL |
| (B) | SNLI | MLM | 11.22 | 19.01 | 1.080 | 0.547 | 13.78 | 74.68 | 4.014 | 1.876 |
| | | MRF$_L$ | 13.39 | 71.44 | 0.433 | 0.267 | 23.45 | 13568.17 | 1.543 | 0.607 |
| | | MRF | 12.30 | 21.65 | 0.658 | 0.179 | 18.35 | 126.05 | 1.967 | 0.366 |
| | | HCB | 12.51 | 22.62 | 0.593 | 0.168 | 17.71 | 589.02 | 2.099 | 0.416 |
| | | AG | 10.76 | 12.68 | 0.007 | 0.085 | 13.26 | 21.59 | 0.018 | 0.181 |
| | XSUM | MLM | 4.88 | 6.12 | 0.404 | 0.227 | 4.910 | 39.33 | 4.381 | 2.128 |
| | | MRF$_L$ | 5.17 | 9.12 | 0.148 | 0.085 | 6.55 | 2209.94 | 1.561 | 0.383 |
| | | MRF | 5.00 | 6.23 | 0.262 | 0.049 | 5.53 | 47.62 | 2.242 | 0.185 |
| | | HCB | 5.08 | 6.21 | 0.256 | 0.052 | 6.46 | 174.32 | 2.681 | 0.328 |
| | | AG | 5.00 | 5.29 | 0.003 | 0.044 | 5.27 | 8.42 | 0.016 | 0.143 |
| (L) | SNLI | MLM | 9.50 | 18.57 | 1.374 | 0.787 | 10.42 | 104.12 | 4.582 | 2.463 |
| | | MRF$_L$ | 11.52 | 76.23 | 0.449 | 0.276 | 15.43 | 8536.92 | 1.470 | 0.543 |
| | | MRF | 10.57 | 19.54 | 0.723 | 0.193 | 13.07 | 93.33 | 1.992 | 0.359 |
| | | HCB | 10.71 | 20.70 | 0.797 | 0.215 | 14.43 | 458.25 | 2.563 | 0.552 |
| | | AG | 8.57 | 10.11 | 0.007 | 0.097 | 9.64 | 15.64 | 0.019 | 0.173 |
| | XSUM | MLM | 3.80 | 5.67 | 0.530 | 0.413 | 3.91 | 103.86 | 5.046 | 3.276 |
| | | MRF$_L$ | 3.94 | 7.06 | 0.156 | 0.068 | 4.62 | 1328.20 | 1.441 | 0.290 |
| | | MRF | 3.87 | 4.94 | 0.322 | 0.036 | 4.16 | 36.66 | 2.258 | 0.145 |
| | | HCB | 3.91 | 5.14 | 0.346 | 0.059 | 5.67 | 164.15 | 2.954 | 0.400 |
| | | AG | 3.88 | 4.13 | 0.003 | 0.042 | 4.21 | 6.62 | 0.016 | 0.126 |

**Table 1:** Comparison of MRF, HCB and AG constructions on randomly sampled SNLI (Bowman et al., 2015) sentences and XSUM (Narayan et al., 2018) summaries. We apply the constructions to two MLMs: BERT$_{BASE}$ ((B)) and BERT$_{LARGE}$ ((L)). We consider both masking tokens uniformly at random (Random pairs) and masking adjacent tokens uniformly at random (Contiguous pairs). PPL metrics measure the quality of the models and KL metrics measure their faithfulness to the MLM's unary conditionals. For all metrics, lower is better.

**Results.** The results are shown in Tab. 1. Comparing the MRF and MRF$_L$ (i.e., the MRF using logits), the former consistently outperforms the latter, indicating that using the raw logits generally results in a worse language model.[7] Comparing the MRFs to MLM, we see that the unary perplexity (U-PPL) of the MLM is lower than those of the MRFs, and that the difference is most pronounced in the contiguous masking case. More surprisingly, we see that the pairwise perplexity (P-PPL) is often (much) higher than the MLM's, even though the MLM makes unrealistic conditional independence assumptions. These results indicate that the derived MRFs are in general worse probabilistic models of language for unary/pairwise tokens (except with an MRF derived from BERT$_{LARGE}$ in the contiguous pair setting). Finally, we also find that the MRFs' unary conditionals are not faithful to those of the MRFs based on the KL measures (A-KL, G-KL).

This example from SNLI qualitatively illustrates a case where both the unary and pairwise perplexities from the MRF underperforms the MLM: "The [MASK]$_1$ [MASK]$_2$ at the casino", where the tokens "man is" are masked. In this case, both MRFs assign virtually zero probability mass to the correct tokens, while the MLM assigns orders of magnitude more (around $0.2\%$ of the mass of the joint). Upon inspection, this arises because $q_{2|1,\overline{S}}(\text{is} \mid \text{man}) \approx 0.02$ and $q_{1|2,\overline{S}}(\text{man} \mid \text{is}) \approx 2 \times 10^{-5}$, which makes the numerator of $q^{MRF}_{1,2|\overline{S}}(\text{man}, \text{is})$ be $\approx 0$. The MRF could still assign high probability to this pair if the denominator is also $\approx 0$, but in this case we have $q_{2|1,\overline{S}}(\text{was} \mid \text{man}) \approx 0.33$ and $q_{1|2,\overline{S}}(\text{man} \mid \text{was}) \approx 0.03$, which makes the denominator well above 0. This causes the completion "man is" to have disproportionately little mass in the joint compared other to combinations ("man was") that were ascribed more mass by BERT's unary conditionals.

## 3.2 TOWARDS FAITHFUL PAIRWISE DISTRIBUTIONS

While the MRF probabilistic model surmounts the conditional independence limitation of MLMs, based on the above results, this comes at the expense of a typically worse probabilistic model overall. Since we ultimately wish to address the conditional independence limitation *in order to* obtain a better probabilistic model, this raises the question: Are there other ways of using the unary distributions (which perform well) to construct a pairwise distribution that does not make a conditional indepen-

---

[7]This gives additional support to the findings of Goyal et al. (2022), who found that MCMC sampling from MRF resulted in sentences with better perplexity than sentences sampled from the MRF$_L$ as measured by GPT2.

dence assumption *and* leads to a better probabilistic model of language? We study two alternative approaches for constructing joint distributions from the unary conditionals of an MLM.

**Hammersley–Clifford–Besag construction.** The Hammersley–Clifford–Besag theorem (HCB; Besag, 1974) provides a way of reconstructing a joint distribution $p(\cdot)$ from its unary conditionals. It states that given a pivot point $\mathbf{w}' = (w'_1, \ldots, w'_T) \in \mathcal{V}^T$, the probability of some $\mathbf{w} \in \mathcal{V}^T$ under the joint distribution $p(\cdot)$ is given by:

$$p(\mathbf{w}) \propto \prod_{t \in [T]} \frac{p_{t|\bar{t}}(w_t \mid \mathbf{w}_{>t}, \mathbf{w}'_{<t})}{p_{t|\bar{t}}(w'_t \mid \mathbf{w}_{>t}, \mathbf{w}'_{<t})} \tag{2}$$

where $\mathbf{w}_{<t} \triangleq (w_1, \ldots, w_{t-1})$, and similarly $\mathbf{w}_{>t} \triangleq (w_{t+1}, \ldots, w_T)$. Importantly, unlike the MRF approach, if the unary conditionals of the MLM *are* compatible (i.e., they are the unary conditionals of some strictly positive joint), then HCB will recover that joint, irrespective of the choice of pivot.

As with the MRF, computing the HCB-implied joint for an MLM would require $\mathcal{O}((T-1)|\mathcal{V}|^{T-1})$ forward passes through the MLM. That said, if we restrict our attention to only two position $S = \{a, b\} \subset [T]$ as before, we can use eq. (2) to construct a distribution over those two positions in $|\mathcal{V}| + 1$ forward passes:

$$q_{a,b|\overline{S}}^{\text{HCB}}(w_a, w_b \mid \mathbf{w}_{\overline{S}}) \propto \frac{q_{a|b,\overline{S}}(w_a \mid w_b, \mathbf{w}_{\overline{S}})}{q_{a|b,\overline{S}}(w'_a \mid w_b, \mathbf{w}_{\overline{S}})} q_{b|a,\overline{S}}(w_b \mid w'_a, \mathbf{w}_{\overline{S}})$$

where $(w'_a, w'_b) \in \mathcal{V}^2$ is some pivot and $\mathbf{w}_{\overline{S}}$ is a context.

**Arnold–Gohkale construction.** One way to frame the faithfulness objective is to find a joint distribution whose unary conditionals have smallest KL to the unary conditionals of the MLM. Since this is intractable for arbitrary joint constructions, we focus on the case of pairwise joints, just as in the MRF and HCB cases. For any pair positions $S = \{a, b\}$ and a context $\mathbf{w}_{\overline{S}}$, we directly optimize for a joint distribution whose unary conditional distributions are faithful:

$$q^{\text{AG}} = \arg\min_{\mu} \sum_{w_a \in \mathcal{V}} D_{\text{KL}}(q_{b|a,\overline{S}}(\cdot \mid w_a, \mathbf{w}_{\overline{S}}) \,\|\, \mu_{b|a,\overline{S}}(\cdot \mid w_a, \mathbf{w}_{\overline{S}}))$$

$$+ \sum_{w_b \in \mathcal{V}} D_{\text{KL}}(q_{a|b,\overline{S}}(\cdot \mid w_b, \mathbf{w}_{\overline{S}}) \,\|\, \mu_{a|b,\overline{S}}(\cdot \mid w_b, \mathbf{w}_{\overline{S}})).$$

Arnold & Gokhale (AG; 1998) study this minimization problem, and provide an effective iterative algorithm for it. The algorithm initializes the starting pairwise distribution $q_{a,b|\overline{S}}^{\text{AG(1)}}(\cdot, \cdot \mid \mathbf{w}_{\overline{S}})$ to be uniform, and makes the following updates until convergence,

$$q_{a,b|\overline{S}}^{\text{AG(t+1)}}(w_a, w_b \mid \mathbf{w}_{\overline{S}}) \propto \frac{q_{a|b,\overline{S}}(w_a \mid w_b, \mathbf{w}_{\overline{S}}) + q_{b|a,\overline{S}}(w_b \mid w_a, \mathbf{w}_{\overline{S}})}{\left(q_{a|\overline{S}}^{\text{AG(t)}}(w_a \mid \mathbf{w}_{\overline{S}})\right)^{-1} + \left(q_{b|\overline{S}}^{\text{AG(t)}}(w_b \mid \mathbf{w}_{\overline{S}})\right)^{-1}}. \tag{3}$$

Obtaining the MLM unary conditionals to be able to run this algorithm requires $2|\mathcal{V}|$ forward passes through the MLM, as in the MRF case.

**Remark.** The HCB construction implicitly assumes compatibility of unary distributions when deriving the joint distribution. On the other hand, the AG construction assumes that the unary distributions themselves are *in*compatible with one another (i.e., there is *no* joint distribution whose set of unary distributions is exactly equal to those of the MLM's; Arnold & Press, 1989), and instead finds a joint distribution that is *near* faithful. Of course, the exact compatibility of learned unary conditionals is unlikely to occur in practice. However, we might again appeal to the rich parameterization of contemporary MLMs to argue that large-scale training results in learned unary conditionals that are close enough to the true unary conditionals (which are by definition compatible). Since both HCB and AG should yield the same exact joint if the unary distributions are compatible, these experiments can assess the extent to which MLMs actually learn compatible unary distributions.

**Experimental setup.** We evaluate the quality of the HCB and AG constructions using the same experimental setup as in the MRF experiments (i.e., same sentences/masks). For the AG joint, we run $t = 50$ steps of the iterative process in eq. (3), which was enough for convergence. For the HCB joint, we pick a pivot using the mode of the pairwise joint of the MLM. (We did not find HCB to be too sensitive to the pivot in preliminary experiments.)

**Results.** The results are shown in Tab. 2. HCB obtains roughly comparable performance to MRF in the random pair masking case. In the contiguous pair case, it exhibits similar failure modes as the MRF in producing extremely high pairwise perplexity (P-PPL) values. The faithfulness metrics are similar to the MRF's. The AG approach, on the other hand, outperforms the $\text{MRF}_L$, MRF and HCB approaches for all metrics. This is most evident in the contiguous masking case, where AG attains lower pairwise perplexity than all models, including the MLM itself. In some cases, we find that the AG model even outperforms the MLM in terms of unary perplexity, which suggests its unary conditionals are better than the MLM's unary conditionals. This is remarkable since the unary conditionals of the MLM were *trained* to approximate the unary conditionals of language, and they do not make any conditional independence assumptions. This suggests that the procedure to construct AG may have a regularizing effect that occasionally leads to a slightly improved probabilistic model of language. The AG's KL measures is substantially lower than the other models, though this is not surprising since AG is trained to optimize that objective. More interestingly, we see that AG's gold KL (G-KL) tends to be on par with the other models, which suggests that it is still not faithful to the MLM in the contexts that are most likely to arise, and moreover confirms that the MLM's unary conditionals themselves are incompatible (Arnold & Press, 1989).

## 4 Learning to be Faithful via Interaction layers

The effectiveness of the AG approach suggests that modeling the dependencies between masked tokens in a sentence should yield an improved probabilistic model of language. However, the above approaches are expensive to run, requiring $\mathcal{O}(|\mathcal{V}|)$ forward passes of the MLM. The MLM, on the other hand, only requires a single forward pass but makes a conditional independence assumption. Can we derive an efficient approach that models pairwise dependencies but only requires a single MLM pass? In this section we study an approach for learning an efficient feed-forward neural network $f$, which we call an **interaction layer**, that learns to adjust the output of the MLM such that the derived model is faithful to the MLM's unary conditionals.

Concretely, suppose we want to build a pairwise (near) faithful model over two positions $S = \{a, b\}$ given a fixed context $\mathbf{w}_{\overline{S}}$. We define a new model that does not make any independence assumptions between the two masked tokens:

$$q^{\text{INT}}_{a,b|\overline{S}}(w_a, w_b \mid \mathbf{w}_{\overline{S}}; \boldsymbol{\theta}) \propto q_{a|\overline{S}}(w_a \mid \mathbf{w}_{\overline{S}}) \, q_{b|\overline{S}}(w_b \mid \mathbf{w}_{\overline{S}}) \, f_{\boldsymbol{\theta}}(w_a, w_b, a, b, \mathbf{w}_{\overline{S}}).$$

This approach allows the new model to exploit the rich linguistic information encoded by the pairwise conditionals of the MLM, without inheriting their conditional independence assumption. Specifically, our function $f$ computes the adjustment factor as:

$$f_{\boldsymbol{\theta}}(w_a, w_b, a, b, \mathbf{w}_{\overline{S}}) \triangleq \exp\left(s \times \left(g_{\boldsymbol{\theta}}(a, w_b, b, \mathbf{w}_{\overline{S}})^\top \text{emb}(w_a) + g_{\boldsymbol{\theta}}(b, w_a, a, \mathbf{w}_{\overline{S}})^\top \text{emb}(w_b)\right)\right)$$

$$g_{\boldsymbol{\theta}}(a, w_b, b, \mathbf{w}_{\overline{S}}) \triangleq \mathbf{Z}\sigma(\mathbf{U}\text{repr}(a; \mathbf{w}_{\overline{S}}) + \mathbf{V}\text{emb}(w_b) + \mathbf{W}\text{pos}(b) + \mathbf{y}) + \mathbf{z}$$

where $\sigma$ is a PReLU (He et al., 2015) activation function, $\text{emb}(w)$ returns the MLM's (word) embedding of a token $w$, $\text{pos}(x)$ returns the positional embedding of the MLM for position $x$, $\text{repr}(x; \mathbf{w}_{\overline{S}})$ is the MLMs contextualized (i.e, final-layer) representation at position $x$ we input the sequence $\mathbf{w}_{\overline{S}}$, and $\boldsymbol{\theta} = \{\mathbf{U}, \mathbf{V}, \mathbf{W}, \mathbf{Z}, \mathbf{y}, \mathbf{z}, s\}$ are learnable parameters. The intuition behind this design is that the function $g(\cdot)$ takes as input the MLM's contextualized representation at the current position (e.g., $a$) and adjusts it according to what would happen if we knew what the token at the other position (e.g., $b$) would be; using these adjusted representations, we then compute an exponentiated dot product with the word embeddings, much like how the final-layer softmax of the MLM is an exponentiated dot product between contextualized representations and the word embeddings.

While the above requires only a single pass of the MLM, $\mathcal{O}(|\mathcal{V}|^2)$ passes of $f$ are needed to evaluate every element of the joint. However, by the Zipfian nature of language, we know this joint should be sparse. For this reason, we propose a training scheme wherein we use $f$ to obtain a *restricted* joint distribution over $K^2$ elements, treating all elements that fall outside that set as having zero probability. In practice, we construct this set of $K^2$ elements by doing a forward pass through the original MLM, obtaining the top-$K$ most likely tokens at each of the two positions, and then taking their Cartesian product.

To train the interaction layer, we minimize the KL divergence between the pairwise conditional implied by AG and the pairwise conditional implied by the interaction layer. Formally, given a dataset of masked sentences $\mathcal{D} = \{(w_a^{(n)}, w_b^{(n)}, \mathbf{w}_{\overline{S}}^{(n)})\}_{n=1}^N$, we optimize the following objective

(the MLM's parameters remain fixed):

$$\arg\min_{\boldsymbol{\theta}} \sum_{n=1}^{N} D_{\text{KL}}(q_{a,b|\overline{S}}^{\text{AG}}(\cdot,\cdot \mid \mathbf{w}_{\overline{S}}^{(n)}) \,||\, q_{a,b|\overline{S}}^{\text{INT}}(\cdot,\cdot \mid \mathbf{w}_{\overline{S}}^{(n)}; \boldsymbol{\theta})).$$

Since we do not normalize over the full vocabulary, we can no longer obtain perplexities. Thus for these experiments we instead resort to standard ranking-based measures. Recall at $k$ (**R@$k$**) computes how often our interaction layer placed original tokens in the sentence (i.e., the ones that were masked) among the top $k$ elements of the joint, viz.,

$$\frac{1}{N} \sum_{n=1}^{N} \mathbb{1}\{\texttt{rank}(w_a^{(n)}, w_b^{(n)}; q_{a,b|\overline{S}}(\cdot \mid \mathbf{w}_{\overline{S}}^{(n)}) \geq k\}.$$

We compute this for $k \in \{1, 5, 10\}$, where $\texttt{rank}(x, y; \mu)$ returns the rank of $(x, y)$ in the joint $\mu$. We also compute mean reciprocal rank (**MRR**):

$$\frac{1}{N} \sum_{n=1}^{N} \left( \texttt{rank}(w_a^{(n)}, w_b^{(n)}; q_{a,b|\overline{S}}(\cdot \mid \mathbf{w}_{\overline{S}}^{(n)})) \right)^{-1}.$$

Note that if the correct tokens $(w_a^{(n)}, w_b^{(n)})$ are not in the restricted joint (i.e., they were assigned zero probability mass), then we take its reciprocal rank to be zero.

**Experimental setup.** We evaluate on the same datasets as before. For our training dataset, we use examples from Wikipedia. We train the interaction layers for 50K steps using the Adam optimizer (Kingma & Ba, 2015), with a learning rate of $0.001$ and a batch size of $64$. We initialize the learnable scalar $s \approx 0$, so that, initially, the joint is only adjusted in a minor way. In preliminary experiments, we found that using the pairwise (unnormalized) logits of the MLM instead of the (normalized) pairwise conditionals worked better. We also lightly explored other variants of the architecture described above (e.g., different ways of combining the embeddings), but found that the design given above worked well enough while being efficient. Finally, since previous experiments indicated that the potential for performance gains over the pairwise conditionals of MLM was modest when masking tokens uniformly at random, we opted to focus solely on the case of masking adjacent tokens (i.e., the contiguous pairs setting).

**Results.** The results shown in Tab. 2 suggest that the interaction layer (INT) tends to outperform the MLM, but underperform other constructions. That said, considering that the joint induced by the interaction layer is much cheaper to compute (i.e., $\mathcal{O}(1)$ MLM forward passes instead of $\mathcal{O}(K^2)$ passes), this positions the interaction layer as a viable alternative to using the MLM's pairwise conditionals. Figure 1 shows that optimizing towards the AG joint (top) improves MRR (bottom).

## 5 DISCUSSION

Our study illuminates the deficiencies of the MRF approach and applies statistically-motivated approaches to craft more performant probabilistic models. However, it is admittedly not clear how these insights can immediately be applied to improve downstream NLP tasks. We focused on models over pairwise tokens in order to avoid sampling and work with exact distributions for the various approaches (MRF, HCB, AG). However this limits the generality of our approach (e.g., we cannot score full sentences). We nonetheless believe that our empirical study is interesting on its own and further suggests new paths for developing efficient, conditionally dependent, and faithful MLMs. For example, one could also use the interaction layer to train against the target gold tokens instead of just conditioning on gold tokens to obtain the unary conditionals with, for example, noise contrastive estimation (Gutmann & Hyvärinen, 2010). Such MLMs trained at scale could provide an alternative to autoregressive language models and provide new functionalities (e.g., controllable editing of sentences).

## 6 RELATED WORK

**Probabilistic interpretations of MLMs.** In one of the earliest works about sampling from MLMs, Wang & Cho (2019) propose to use unary conditionals to sample sentences. Recently Yamakoshi et al. (2022) highlight that, while Wang & Cho's (2019) approach only constitutes a pseudo-Gibbs sampler, the act of re-sampling positions uniformly at random guarantees that the resulting Markov

| | Dataset | Scheme | Contiguous pairs | | | |
| | | | R@1 | R@5 | R@10 | MRR |
|---|---|---|---|---|---|---|
| | | MLM | 0.117 | 0.260 | 0.305 | 0.184 |
| | | $MRF_L$ | 0.175 | 0.379 | 0.458 | 0.271 |
| | SNLI | MRF | 0.140 | 0.333 | 0.424 | 0.235 |
| | | HCB | 0.167 | 0.364 | 0.451 | 0.264 |
| | | AG | 0.219 | 0.416 | 0.473 | 0.311 |
| Ⓑ | | INT | 0.145 | 0.299 | 0.363 | 0.216 |
| | | MLM | 0.183 | 0.342 | 0.401 | 0.259 |
| | | $MRF_L$ | 0.339 | 0.579 | 0.660 | 0.446 |
| | XSUM | MRF | 0.304 | 0.523 | 0.615 | 0.407 |
| | | HCB | 0.317 | 0.550 | 0.625 | 0.425 |
| | | AG | 0.379 | 0.593 | 0.670 | 0.479 |
| | | INT | 0.233 | 0.400 | 0.465 | 0.316 |
| | | MLM | 0.110 | 0.235 | 0.288 | 0.172 |
| | | $MRF_L$ | 0.231 | 0.419 | 0.518 | 0.324 |
| | SNLI | MRF | 0.170 | 0.374 | 0.474 | 0.270 |
| | | HCB | 0.204 | 0.396 | 0.484 | 0.298 |
| | | AG | 0.234 | 0.442 | 0.530 | 0.334 |
| Ⓛ | | INT | 0.127 | 0.265 | 0.328 | 0.196 |
| | | MLM | 0.151 | 0.288 | 0.343 | 0.215 |
| | | $MRF_L$ | 0.399 | 0.639 | 0.721 | 0.508 |
| | XSUM | MRF | 0.352 | 0.592 | 0.689 | 0.462 |
| | | HCB | 0.360 | 0.568 | 0.659 | 0.459 |
| | | AG | 0.414 | 0.647 | 0.723 | 0.518 |
| | | INT | 0.162 | 0.324 | 0.402 | 0.242 |

**Table 2:** Comparison of interaction layer (INT), when applied to both BERT$_{\text{BASE}}$ (Ⓑ) and BERT$_{\text{LARGE}}$ (Ⓛ), to previous joint constructions.

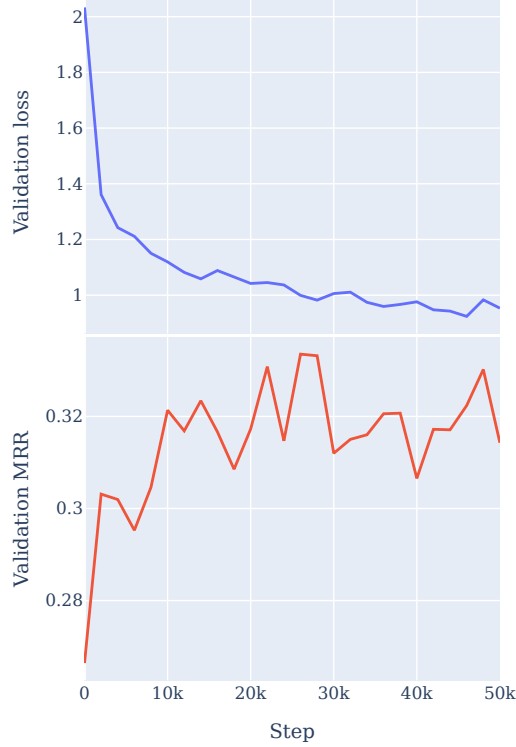

**Figure 1:** Validation loss for optimizing towards AG joint (top) and mean reciprocal rank (bottom) for BERT$_{\text{BASE}}$ interaction layer during training.

chain has a unique, stationary distribution (Bengio et al., 2013; 2014). In contrast to sampling directly from BERT, Goyal et al. (2022) propose deriving an MRF from the unary conditionals learned by BERT, and sample from this via Metropolis-Hastings. Ghazvininejad et al. (2019) and Savinov et al. (2022) use conditional MLMs for fast decoding of translation models. Our interaction layer is related to residual energy networks (Deng et al., 2020), which also learn to modify the output from an existing model.

**Compatible distributions.** The statistics community has long studied the problem of assessing the compatibility of a set of conditionals (Arnold & Press, 1989; Gelman & Speed, 1993; Wang & Kuo, 2010; Song et al., 2010). Arnold & Gokhale (1998) and Arnold et al. (2002) explore algorithms for reconstructing near-compatible joints from incompatible conditionals, which we leverage in our work. Besag (1974) also explores this problem, and defines a procedure (viz., eq. (2)) for doing so when the joint distribution is strictly positive and the conditionals are compatible. Lowd (2012) apply a version of HCB to derive Markov networks from incompatible dependency networks (Heckerman et al., 2000). Pseudo-gibbs sampling, which refers to sampling from incompatible (or near compatible) conditionals, has also been studied from both empirical and theoretical perspectives (Chen et al., 2011; Chen & Ip, 2015; Kuo & Wang, 2019).

## 7 CONCLUSION

In this paper we studied pairwise MRFs derived from unary conditionals of MLMs and empirically observed them to not only be worse language models but also have unary distribution that do not match the original MLM's. We then studied two statistically motivated approaches for deriving more faithful MRFs, and found that the iterative optimization approach, which identifies a joint that is more faithful to the unary conditionals of the original MLM, performs well. Finally, we experimented with amortizing the optimization algorithm via a learned feed-forward layer to derive a conditionally dependent language model.

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

## A    UNFAITHFUL MRFS

Here we show that even if the unary conditionals used in the MRF construction are compatible (Arnold & Press, 1989), the unary conditionals of the probabilistic model implied by the MRF construction can deviate (in the KL sense) from the true conditionals. This is important because (i) it suggests that we might do better (at least in terms unary PPL) by simply sticking to the conditionals learned by MLM, and (ii) this is not the case for either the HCB or the AG models, i.e., if we started with the correct conditionals, HCB and AG's joint would be compatible with the MLM.

To see this, suppose we defined our MRF using the unary conditionals of the true data distribution, $p$. Further, for simplicity, suppose $T = 2$. Hence, our MRF has the form:

$$q_{1,2}(w_1, w_2) \propto p_{1|2}(w_1 \mid w_2)\, p_{2|1}(w_2 \mid w_1)$$

Then for some $w_2 \in \mathcal{V}$, we have:

$$q_{1|2}(w_1 \mid w_2) = \frac{p_{1|2}(w_1 \mid w_2)\, p_{2|1}(w_2 \mid w_1)}{\sum_{w' \in \mathcal{V}} p_{1|2}(w' \mid w_2)\, p_{2|1}(w_2 \mid w')}$$

Now, consider the KL between the true unary conditionals and the MRF unary conditionals:

$$D_{\mathrm{KL}}(p_{1|2}(\cdot \mid w_2) \,\|\, q_{1|2}(\cdot \mid w_2)) = \sum_{w \in \mathcal{V}} p_{1|2}(w \mid w_2) \log \frac{p_{1|2}(w \mid w_2)}{q_{1|2}(w \mid w_2)}$$

$$= \sum_{w \in \mathcal{V}} p_{1|2}(w \mid w_2) \log \frac{\sum_{w' \in \mathcal{V}} p_{1|2}(w' \mid w_2)\, p_{2|1}(w_2 \mid w')}{p_{2|1}(w_2 \mid w)}$$

$$= \log \mathbb{E}_{w \sim p_{1|2}(\cdot|w_2)}[p_{2|1}(w_2 \mid w)] - \mathbb{E}_{w \sim p_{1|2}(\cdot|w_2)}[\log p_{2|1}(w_2 \mid w)]$$

This term is the Jensen gap, and in general it can be non-zero. To see this, suppose $\mathcal{V} = \{a, b\}$ and consider the joint

$$p_{1,2}(w_1, w_2) = \begin{cases} \frac{97}{100} & w_1, w_2 = a \\ \frac{1}{100} & \text{otherwise} \end{cases}$$

with corresponding conditionals $p_{2|1}(x \mid b) = p_{1|2}(x \mid b) = \frac{1}{2}$ for all $x \in \mathcal{V}$ and

$$p_{2|1}(x \mid a) = p_{1|2}(x \mid a) = \begin{cases} \frac{97}{98} & x = a \\ \frac{1}{98} & x = b \end{cases}$$

Now, take $w_2 = b$. We then have

$$D_{\mathrm{KL}}(p_{1|2}(\cdot \mid b) \,\|\, q_{1|2}(\cdot \mid b))$$

$$= \log \mathbb{E}_{w \sim p_{1|2}(\cdot|b)}[p_{2|1}(b \mid w)] - \mathbb{E}_{w \sim p_{1|2}(\cdot|b)}[\log p_{2|1}(b \mid w)]$$

$$= \log \left( \frac{1}{2} \left( \frac{1}{98} + \frac{1}{2} \right) \right) - \frac{1}{2} \left( \log \frac{1}{98} + \log \frac{1}{2} \right)$$

$$= \log \left( \frac{1}{196} + \frac{1}{4} \right) - \frac{1}{2} \left( \log \frac{1}{196} \right) \approx 1.27$$

which demonstrates that the KL can be non-zero.

