# OpenReview forum: "Towards Conditionally Dependent Masked Language Models"
_ICLR.cc/2023/Conference — Submitted to ICLR 2023_

### Official Review · Reviewer_fxig · 2022-10-24

**Confidence:** 3
**Correctness:** 4
**Technical Novelty And Significance:** 2
**Empirical Novelty And Significance:** 2
**Recommendation:** 5

**Clarity, Quality, Novelty And Reproducibility:**

The writing is clear and well structured. The studied problem is important and relevant to the conference.


**Strength And Weaknesses:**

Strengths
- the paper identifies that previous attempts at interpreting MLMs as MRFs may not be faithful, in the sense that unary conditionals may not match.
- the paper improves the 'faithfulness' of the joint distribution via a learned interaction layer which allows modeling the joint with just one evaluation pass (by using a mixing function and the independent unary conditionals).

Weaknesses
- the contribution only focus on pairwise MRFs, which seems incremental. Why not generalize to cliques of size k for some small k, or arbitrary MRFs with low width?


Question
- I had a question throughout the paper, not about this particular paper but about the whole general direction of interpreting MLM as MRFs. It seems that we are jumping through a lot of hoops to define a MRF, find joint distributions with faithful conditionals, run expensive inference, etc.... Can't we simply use the MLM unary conditionals directly in an autoregressive way and avoid all of this trouble? E.g. see https://arxiv.org/abs/2110.02037 https://arxiv.org/abs/2205.13554

**Summary Of The Paper:**

This paper examines the perspective of interpreting masked language models (MLMs) as a generative model, where the unmasking procedure corresponds to predicting unary conditionals. The paper notes that the different unmasked tokens are predicted in a conditionally independent way, and focuses previous attempts that instead interpret the MLM as a Markov random field to overcome the conditional independence limitations.

First the paper shows the limitations of MRF methods by showing that pairwise MRFs (when two tokens are masked) can be worse than the naive conditionally independent model. Second, they propose an iterative optimization problem, with amortization for computational efficiency, that improves MLM for scoring pairwise tokens.

**Summary Of The Review:**

The problem of generation with MLMs is important and has been studied before. This paper shows improvements with pairwise MRFs and has good writing and execution. The scope of the improvement however, may not be that large.

---

> ### Author Response · Authors · 2022-11-19
> **Response**
>
> Thank you for your review!
>
> - **the contribution only focus on pairwise MRFs, which seems incremental.** We chose to focus on the pairwise case since extending our results to the case of more than two tokens would require us to compute metrics approximately (e.g., paraphrasing from the comment above: if we mask 3 tokens, then merely storing the joint would require ~100TB, which is infeasible). This in turn would add additional noise to our comparisons, which we sought to avoid. Hence, if we care about exact comparisons, then k=2 is just about as large (clique-wise) as we can go.
>
> - **I had a question throughout the paper...**  Yes—there certainly are other conceptually simpler approaches to auto-regressive generation. Indeed Wang & Cho (2019) explored decoding left-to-right auto-regressively using MLM conditionals, but found it to be empirically ineffective compared to using pseudo-Gibbs sampling. More generally, to the best of our knowledge, an any-order autoregressive approach either (i) doesn’t define a single joint distribution (i.e., the order in which we generate tokens may effectively lead to different joints) or (ii) _implicitly_ defines a joint, but manipulating it is intractable (e.g., the joint is a mixture of the joints implied by each ordering). In turn, inference in this regime become approximate, which was the problem we were seeking to avoid initially by defining a probabilistic model.

---

### Official Review · Reviewer_zRNa · 2022-10-24

**Confidence:** 2
**Correctness:** 3
**Technical Novelty And Significance:** 3
**Empirical Novelty And Significance:** 2
**Recommendation:** 6

**Clarity, Quality, Novelty And Reproducibility:**

The paper introduces so-called pairwise MRFs in order to define language models for MLMs when pairs of words are masked. From an NLP perspective, I do not see for which applications it could be useful to mask pairs of words. For applications, the more interesting case should be when the two tokens are contiguous, i.e. the case for which the independency condition is false. Moreover, if such applications do exist, why would it be useful to consider them as language models for these applications. Let us suppose that pairwise MRFs are useful, now is the question why the unary distributions they defined should coincide with the distributions of the original MLM.

I think the paper could be made clearer: it is not always easy to see which MLM you are speaking of (pairwise MLM, original MLM, parent MLM, ...); it should also made clear at the beginning that a pairwise MLM is not trained with masked pairs but that you consider "classical" MLMs. This being said, the study is well designed from both a theoretical and experimental perspective. It does not come as a surprise that the AG model performs better given that the model is optimized for a joint distribution which is faithful to the original MLM. The experimental results for the algorithmic approach are promising.

**Strength And Weaknesses:**

Pros

* Showing that distributions defined with two masked tokens do not define unary distributions close to that of the original MLM
* Proposal of statistically founded methods to answer this problem
* Proposal of an algorithmic solution to give a proxy

Cons

* The motivations are not clear at least to me

**Summary Of The Paper:**

The paper studies how a masked language model (MLM) can be used as a traditional language model. The authors consider the case of two masked tokens and they proposed pairwise Markov random fields (MRFs). They experimentally show that pairwise MRFs are worse probabilistic models of language from a perplexity standpoint. They studied two formal approaches for deriving better language models and one algorithmic solution using a feed-forward neural layer.

**Summary Of The Review:**

The paper contains a statiscally funded study completed by interesting experiments. The motivations should be made explicit. For now, I am not convinced by the usefulness of the study at least from an NLP perspective.

---

> ### Author Response · Authors · 2022-11-19
> **Response**
>
> Thank you for your review!
>
> Our focus in this paper was to study different methods of using the conditionals learnt by MLMs to define a probabilistic model of language. Since our focus was in comparing the joints derived by these models, we focus on the case of pairwise coherent MLMs, i.e., when only two tokens are masked, and so the distribution is over those two tokens. This is important, since evaluating the quality of these models is intractable if we consider more than two tokens (e.g., if the number of tokens V in our vocabulary is 30K, then storing a pairwise joint requires ~4GB, but storing a triplewise joint would require ~100TB). If we wanted to mask more than 2 tokens, we would have to rely on approximations to evaluate the quality of the joints, which would add stochasticity to our comparisons. This makes such extensions to more than two tokens orthogonal to our work.
>
> Thank you for your feedback! We will make sure to clarify which MLM we refer to at different points in the text, and make it clearer what we mean by a pairwise MLM (in contrast to the base classical MLM).

---

### Official Review · Reviewer_cQUX · 2022-10-27

**Confidence:** 4
**Clarity, Quality, Novelty And Reproducibility:** See summary of the review below.
**Correctness:** 2
**Technical Novelty And Significance:** 2
**Empirical Novelty And Significance:** 2
**Recommendation:** 5

**Strength And Weaknesses:**

See summary of the review below.

**Summary Of The Paper:**

See summary of the review below.

**Summary Of The Review:**

I think this is an interesting paper, but I found the motivation for the approach lacking.

The paper describes various issues in going from MLM parameter estimation objectives, to density estimation over sequences. The paper first describes a number of naive methods (section 3.1.1) that are clearly heuristic, lacking clear guarantees, and which not surprisingly lead to bad performance. The paper then describes prior results from pseudo-likelihood that do lead to well-founded estimators, and give better performance.

This doesn’t feel like a significant enough contribution for ICLR, for two reasons. 1) it’s not surprising at all that using well-founded methods, known a long way back in the pseudo-likelihood literature, give improvements. 2) density estimation of this type is of limited interest, as auto-regressive models do not have these issues; compelling arguments need to be made for why the MRF-based approach is preferable, and how the method might be scaled from pair-wise interactions.


In addition, I think the motivation/discussion in the introduction needs to be clarified, specifically:

“From a probabilistic perspective, masked language models (MLM) make strong independence assumptions”. There is *a lot* riding on the phrase “from a probabilistic perspective” here. I think what the authors are getting at is going from the MLM training criterion to a full density p(x) over sentences x - and how this may be challenging given the way current MLM objectives are set up. That’s ok, but two comments: 1) the original MLM objective was never motivated for density estimation, i.e., estimation of p(x), it was used for pretraining/representation learning. It was always clear that there might be some connection through pseudo-likelhood to density estimation, but the method never depended on that link being made. The MLM method is also completely sound in estimating a particular conditional distribution that arises from masking the input sentence. 2) the authors themselves do not give a particularly compelling method to density estimation themselves, which leaves the motivation on somewhat shaky ground.

The introduction continues in this vain: “such model misspecifications arising from incorrect statistical assumptions may not be catastrophic”. Again, the original MLM objective did not try to derive a model for p(x), so the term “model misspecification” is in my opinion highly misleading here.

“However, we observe that MLMs are increasingly being employed as probabilistic models of language” - presumably by a “probabilistic model of language” you mean a density p(x) over sentences? The authors need to clarify these terms.

I’m concerned that a naive reader will get the impression that MLM methods are fundamentally unsound, and that is just not a correct characterization in my opinion.

---

> ### Author Response · Authors · 2022-11-19
> **Response**
>
> Thank you for your thoughtful review! We fully agree with you that the original MLM was not motivated as a probabilistic model of language. However, as we note in the paper, MLMs *have* been used as implicit probabilistic models (and sometimes effectively so!). Thus, the question of deriving language models (i.e. p(x)) given an MLM is an interesting question, in our opinion.
>
> We respond to some of the more specific points below.
> - **it’s not surprising at all that using well-founded methods, known a long way back in the pseudo-likelihood literature, give improvements.**: As you indicate, the MRF method lacks clear guarantees, whereas the other methods in section 3.2 offer clearer guarantees. However, we don’t think that this is necessarily unsurprising. Current work in treating MLMs probabilistically for the purposes of sampling has advocated in favor of MRF (e.g., Goyal et al., 2021) or using pseudo-Gibbs sampling (e.g., Wang & Cho, 2019). For this reason, we believe that there is value in shining light on these methods, in the hopes that they could ultimately lead to better sampling from MLMs. Moreover, although the methods we explore (Hammersley--Clifford--Besag and the Arnold--Gohkale constructions) go a long way back, revisiting these methods in the context of contemporary MLMs is interesting and novel in our opinion.
>
> - **density estimation of this type is of limited interest, as auto-regressive models do not have these issues...** We do agree that auto-regressive models do not exhibit the issues we point out in the paper. After all, they explicitly define a joint distribution over sentences. However, there has been interest in reinterpreting MLMs as probabilistic models, and in certain situations they can be favored over auto-regressive models: For example, in sampling, Ghazvininejad et al. (2019) find that if one controls for the quality of the generated samples, then MLM sampling is better (in terms of computational efficiency) than auto-regressive sampling, since in MLMs, multitoken sampling can be done in a single forward pass. More generally, we believe that the fact that some work (directly or indirectly) treats MLMs as if they were probabilistic models is what, in our view, motivates this paper.
>
> - **the original MLM objective was never motivated for density estimation, i.e., estimation of p(x), it was used for pretraining/representation learning.** We agree that MLMs, as a representation learning objective, are extremely effective, and that they were not introduced for the purposes of density estimation. However, our point (referenced in “we observe that MLMs are increasingly being employed as probabilistic models of language”) is that notwithstanding the above, MLMs are being interpreted probabilistically, e.g., by the examples given above. Hence, if one is going to use and interpret MLMs probabilistically, then one may as well strive to do it in a more principled manner.
>
> - **you mean a density p(x) over sentences? The authors need to clarify these terms.** Yep! By $p(x)$ we mean a language model in the sense of a pmf over $\mathcal{V}^T$.

---

### Author Response · Authors · 2022-11-19
**General response**

Thank you all for your reviews!

### Motivation
Many reviewers found the motivation and the "so what" of aspect of the paper unclear. Our study was motivated by the observation that despite MLM's being originally introduced in the context of representation learning, they have been quite successful as implicit probabilistic models of language for scoring and sampling sentences. We take this as our starting point and revisit classic methods from statistics to see if these methods could be applied to modern MLMs to derive explicit probabilistic models of language from the original MLM. Our main finding is that out of various MLM-derived models (MRF, HCB, AG), those that are faithful to the original MLM's unary distribution (i.e., the AG approach) performs best as a language model, at least for pairwise tokens. While this does not have clear engineering applications, we nonetheless think that this is an interesting finding, and could motivate development of new classes of MLMs.

### Additional analyses
Also, we performed additional analyses pertaining to how the distance between masked positions affects the quality of the joint. To this end, we plotted the average pairwise negative log-likelihood (PNLL; lower is better) as a function of the (i) token distance between masked tokens and the (ii) syntactic distance between masked tokens. We add links to the four plots below:

- SNLI: https://imgur.com/a/YqJP6sz
- XSUM: https://imgur.com/a/OphIEWm

Note that the black bars denote the number of examples available in each bucket. Also note that a syntactic distance 0 denotes the two masked tokens belong to the same word, whereas a token distance of 0 denotes the two masked tokens are adjacent.

The graphs indicate that the performance improvement is most prominent when masked tokens are close together, confirming the observations in the paper. In this case, AG tends to do best, HCB and MRF tend to do similarly, followed by MRF-L and, finally, the naive, conditionally independent MLM, which follows the trends in the paper.

---

### Decision · Program_Chairs · 2023-01-20

**Decision:**

Reject

**Justification For Why Not Higher Score:**

A clear reviewer consensus emerged during discussion.

**Justification For Why Not Lower Score:**

N/A

**Metareview: Summary, Strengths And Weaknesses:**

This paper discusses limitations of MLMs, interpreted as MRFs, and proposes better alternatives. It received a lukewarm, although not wholly negative, reaction from reviewers. As it was borderline, we discussed it async, the results of said discussion being summarized below. The consensus shifted towards agreeing on rejection, with the point being made that despite the clarity of the writing, the justification for the premise of the paper was not strong enough. To quote one of the reviewers during discussion: "Autoregressive unrolling methods of masked language models already mitigates the conditionally independent assumption. Moreover, the computational limitation of the methodology to pairwise MRFs makes the scope rather narrow." From reading through the comments and discussion, and the rebuttals, I agree with the consensus that was reached. At very least, a better framing of the necessity for this work is desirable, and I hope that by incorporating the feedback offered by reviewers, the authors will have a better result upon resubmission to another conference.

**Summary Of Ac-Reviewer Meeting:**

The discussion was held async via openreview. The sole reviewer initially advocating for borderline acceptance agreed that the reviewer consensus was that interpreting MLMs as MRFs is not a so important problem, and that the study is limited to so-called pairwise MRFs. The reviewers found the rebuttal not compelling in terms of changing their views, as it mostly just restated the points made in the paper rather than offering additional clarification.